# Intravascular Lithotripsy for the Treatment of Stent Underexpansion: The Multicenter IVL-DRAGON Registry

**DOI:** 10.3390/jcm11071779

**Published:** 2022-03-23

**Authors:** Wojciech Wańha, Mariusz Tomaniak, Piotr Wańczura, Jacek Bil, Rafał Januszek, Rafał Wolny, Maksymilian P. Opolski, Łukasz Kuźma, Adam Janas, Tomasz Figatowski, Paweł Gąsior, Marek Milewski, Magda Roleder-Dylewska, Łukasz Lewicki, Jan Kulczycki, Adrian Włodarczak, Brunon Tomasiewicz, Sylwia Iwańczyk, Jerzy Sacha, Łukasz Koltowski, Miłosz Dziarmaga, Miłosz Jaguszewski, Paweł Kralisz, Bartosz Olajossy, Grzegorz Sobieszek, Krzysztof Dyrbuś, Mariusz Łebek, Grzegorz Smolka, Krzysztof Reczuch, Robert J. Gil, Sławomir Dobrzycki, Piotr Kwiatkowski, Marcin Rogala, Mariusz Gąsior, Andrzej Ochała, Janusz Kochman, Adam Witkowski, Maciej Lesiak, Fabrizio D’Ascenzo, Stanisław Bartuś, Wojciech Wojakowski

**Affiliations:** 1Department of Cardiology and Structural Heart Diseases, Medical University of Silesia, 40-055 Katowice, Poland; p.m.gasior@gmail.com (P.G.); marek.milewski92@gmail.com (M.M.); magdaroleder@gmail.com (M.R.-D.); gsmolka@me.com (G.S.); aochala1@gmail.com (A.O.); wojtek.wojakowski@gmail.com (W.W.); 21st Department of Cardiology, Medical University of Warsaw, 02-091 Warszawa, Poland; tomaniak.mariusz@gmail.com (M.T.); lukasz@koltowski.com (Ł.K.); jkochman@wum.edu.pl (J.K.); 3Department of Cardiology, The Ministry of Internal Affairs and Administration Hospital, 35-111 Rzeszów, Poland; p.wanczura@poczta.fm; 4Department of Invasive Cardiology, Centre of Postgraduate Medical Education, 02-507 Warsaw, Poland; biljacek@gmail.com (J.B.); scorpirg@gmail.com (R.J.G.); 5Department of Cardiology, Jagiellonian University Medical College, 31-008 Krakow, Poland; jaanraf@interia.pl (R.J.); mbbartus@gmail.com (S.B.); 6Department of Interventional Cardiology and Angiology, National Institute of Cardiology, 04-628 Warsaw, Poland; rafal.wolny@gmail.com (R.W.); opolski.mp@gmail.com (M.P.O.); witkowski@hbz.pl (A.W.); 7Department of Invasive Cardiology, Medical University of Bialystok, 15-089 Bialystok, Poland; kuzma.lukasz@gmail.com (Ł.K.); paqral@yahoo.com (P.K.); slawek_dobrzycki@yahoo.com (S.D.); 8Faculty of Medicine and Health Science, Andrzej Frycz Modrzewski Kraków University, 30-705 Krakow, Poland; adamjjanas@gmail.com; 9First Department of Cardiology, Medical University of Gdansk, 80-210 Gdansk, Poland; figatowski@gumed.edu.pl (T.F.); jamilosz@gmail.com (M.J.); 10Department of Invasive Cardiology, University Center for Cardiology, 80-211 Gdansk, Poland; luklewicki@gmail.com; 11Department of Cardiology, Miedziowe Centrum Zdrowia, 59-300 Lubin, Poland; jan.jakub.kulczycki@gmail.com (J.K.); wlodarczak.adrian@gmail.com (A.W.); 12Centre for Heart Disease, University Hospital, 50-558 Wroclaw, Poland; b.a.tomasiewicz@gmail.com (B.T.); reczuch@wp.pl (K.R.); 13Department of Heart Disease, Wroclaw Medical University, 50-367 Wroclaw, Poland; 14Department of Cardiology, Poznan University of Medical Sciences, 61-701 Poznan, Poland; syl.iwanczyk@gmail.com (S.I.); maciej.lesiak@skpp.edu.pl (M.L.); 15Department of Cardiology, University Hospital, Institute of Medical Sciences, University of Opole, 45-040 Opole, Poland; sacha@op.pl; 16Department of Cardiology—Intensive Therapy and Internal Diseases, Poznan University of Medical Sciences, 60-355 Poznan, Poland; milosz.dziarmaga@gmail.com; 171st Military Hospital in Lublin, 20-049 Lublin, Poland; bolajossy@gmail.com (B.O.); grzes.bies@interia.pl (G.S.); 18Third Department of Cardiology, Medical University of Silesia, 40-055 Katowice, Poland; k.dyrbus@sccs.pl (K.D.); m.gasior@op.pl (M.G.); 19Upper Silesia Medical Centre, 40-635 Katowice, Poland; mlebek@poczta.onet.pl; 20Department of Cardiology and Internal Diseases, Military Institute of Medicine, 04-141 Warsaw, Poland; pkwiatkowski@wim.mil.pl; 21Terumo, 02-134 Warszawa, Poland; rogmar78@yahoo.co.uk; 22Division of Cardiology, Department of Internal Medicine, Città della Salute e della Scienza, University of Turin, 10124 Turin, Italy; fabrizio.dascenzo@gmail.com

**Keywords:** percutaneous coronary intervention, stent underexpansion, intravascular lithotripsy, calcified lesions

## Abstract

Background: Whereas the efficacy and safety of intravascular lithotripsy (IVL) have been confirmed in de novo calcified coronary lesions, little is known about its utility in treating stent underexpansion. This study aimed to investigate the impact of IVL in treating stent underexpansion. Methods and Results: Consecutive patients with stent underexpansion treated with IVL entered the multicenter IVL-Dragon Registry. The procedural success (primary efficacy endpoint) was defined as a relative stent expansion >80%. Thirty days device-oriented composite endpoint (DOCE) (defined as a composite of cardiac death, target lesion revascularization, or target vessel myocardial infarction) was the secondary endpoint. A total of 62 patients were enrolled. The primary efficacy endpoint was achieved in 72.6% of patients. Both stent underexpansion 58.5% (47.5–69.7) vs. 11.4% (5.8–20.7), *p* < 0.001, and the stenotic area 82.6% (72.4–90.8) vs. 21.5% (11.1–37.2), *p* < 0.001, measured by quantitative coronary angiography improved significantly after IVL. Intravascular imaging confirmed increased stent expansion following IVL from 37.5% (16.0–66.0) to 86.0% (69.2–90.7), *p* < 0.001, by optical coherence tomography and from 57.0% (31.5–77.2) to 89.0% (85.0–92.0), *p* = 0.002, by intravascular ultrasound. Secondary endpoint occurred in one (1.6%) patient caused by cardiac death. There was no target lesion revascularization or target vessel myocardial infarction during the 30-day follow-up. Conclusions: In this real-life, largest-to-date analysis of IVL use to manage underexpanded stent, IVL proved to be an effective and safe method for facilitating stent expansion and increasing luminal gain.

## 1. Introduction

Percutaneous coronary intervention (PCI) with stent implantation is a well-known treatment option for patients with significant coronary stenosis [1]. Adequate stent expansion is the main factor contributing to PCI outcomes, with stent underexpansion being the most potent predictor of in-stent restenosis (ISR) and stent thrombosis [2,3]. The data from the study by Kang et al. [2] showed that 42% of ISR lesions have stent underexpansion. Notably, underexpanded stent lesions have unique characteristics when compared with the de novo atherosclerotic lesions. The plaque in underexpanded stents often consists of heavy and deep calcifications, resistant to dilatation. Additionally, percutaneous treatment of underexpanded stents is challenging and often leads to suboptimal clinical outcomes. Typically, high-pressure balloon inflation remains the first-line treatment of underexpanded stents. However, while this technique is effective in the non-heavily calcified lesion, balloon inflation is often not sufficient for complete stent expansion within massive calcifications, and on the other hand, it carries a risk of artery dissection or perforation [4]. Therefore, several strategies and technologies have emerged over the past decade to improve the management of stent underexpansion [5,6,7]. Amongst them, intravascular lithotripsy (IVL) (Shockwave Medical Inc., Santa Clara, United States) has been introduced. The IVL system produces mechanical shock waves that propagate from the balloon and induce superficial and deep calcium fractures without affecting soft tissue. The efficacy and safety of IVL have been confirmed in de novo calcified lesions in Disrupt CAD I-IV studies [4,8,9,10]. However, the data on IVL utility in the treatment of stent underexpansion remains scarce and is limited to case reports [11,12,13,14]. In addition, to date, there are no clear guidelines regarding which type of method should be preferred during PCI with stent underexpansion. Here, we report the results of the multicenter IVL-Dragon Registry aiming to assess the efficacy and safety of IVL in coronary stent underexpansion.

## 2. Methods

The IVL-Dragon Registry was a large multicenter, retrospective, observational cohort study conducted in high-volume PCI centers. The dataset included patients with stent underexpansion treated with IVL between November 2019 and April 2021. Derivation of the final study cohort is shown in Appendix A. Stent underexpansion was diagnosed by two independent interventional cardiologists at each participating center as a relative stent expansion of <80% (stenosis at MLD divided by average reference lumen area). It was evaluated by in-stent diameter stenosis as assessed by quantitative coronary angiography (QCA). Heavy calcification was defined as calcification within the lesion on both sides of the vessel that the operator assessed on the cine angiographic still frame. The stent underexpansion was treated either during the clinically indicated index procedure or referred for planned IVL because of suboptimal expansion achieved at the end of another procedure. The angiographic, QCA, intravascular ultrasound (IVUS), and optical coherence tomography (OCT) data included in the study were collected, anonymized, and recorded in the central cardiovascular information registry. The patients’ data were fully anonymized in each center, combined into the database, and statistically analyzed together as a single cohort. The patient’s data were protected according to the requirements of Polish law and hospital standard operating procedures. The study was conducted in accordance with the Declaration of Helsinki and was registered at ClinicalTrials.gov (NCT05112250).

### 2.1. Study Device

The Shockwave IVL System with the Shockwave C2 Coronary IVL Catheter is indicated for lithotripsy-enabled, low-pressure balloon dilatation of severely calcified, stenotic coronary arteries. A 6F compatible, rapid-exchange delivery system delivers the balloon over a conventional 0.014′ guidewire. IVL balloon is available in four diameters: 2.5, 3.0, 3.5, and 4.0 mm and in one length of 12 mm. The maximum pulse count is 80 pulses per catheter.

### 2.2. Patient Follow-Up and Study Endpoints

The primary efficacy endpoint was the procedural success, defined as a relative stent expansion >80. The secondary endpoint was freedom from device-oriented composite endpoint (DOCE) (defined as a composite of cardiac death, target lesion revascularization (TLR), target lesion revascularization, (TLR), and target vessel myocardial infarction (MI)) at 30 days. Endpoints were defined according to previously proposed standards [15]. Data regarding long-term outcomes were obtained by phone call or clinical visit as well as from the National Health Fund Service database, and no patient was lost to follow-up.

### 2.3. Procedure

Stent implantation or non-IVL balloon dilatation was made at the operator’s discretion and according to the European Society of Cardiology guidelines for myocardial revascularization [1]. Each patient received a dual antiplatelet therapy for a minimum of 6 months. 

### 2.4. QCA Measurements

QCA was performed offline using a validated Medis Suite XA/QAngio XA 3D software (Medis, Leiden, The Netherlands) to calculate minimum lumen diameter, percentage stenosis at minimum lumen diameter, area of stenosis at minimum lumen diameter, and lesion length. The 3D QCA analyses were analyzed ≥2 angiographic projections separated by 30°.

### 2.5. OCT Image Acquisition

The commercially available ILUMIEN OPTIS PCI Optimization^TM^ System (Abbott, Plymouth, MN, USA) was used for image acquisition. The following parameters were obtained: minimal lumen area, minimal stent area, stent expansion at minimal stent area, maximal calcium angle behind stent, maximum calcium thickness, calcium length behind stent, malapposition, thrombus, stent fracture, tissue protrusion, dissection/intramural hematoma. Stent expansion in OCT image was defined as the minimal stent area divided by the mean of the proximal and distal reference lumen areas [16].

### 2.6. IVUS Image Acquisition

The commercially available IVUS systems, including POLARIS^TM^ with OptiCross^TM^ rotational catheters (Boston Scientific, Marlborough, MA, USA) and Core^TM^ with rotational Refinity and digital EagleEye^TM^ catheters (Philips Volcano, Amsterdam, The Netherlands), were used for image acquisition. The following parameters were obtained: minimal lumen area, minimal stent area, stent expansion at minimal stent area, plaque burden behind stent at minimal lumen area, remodeling index, dissection/intramural hematoma. Stent expansion in IVUS image was defined as the minimal stent area divided by the mean of the proximal and distal reference lumen areas [16].

### 2.7. Statistical Analysis

Continuous data are presented as mean ± standard deviation or median with IQR [Q1–Q3]. Categorical data are expressed as count and percentage. Normal distribution was verified by the Kolmogorov–Smirnov test. Continuous data were compared by the Student *t* test or by Mann–Whitney *U* test, depending on the data distribution. Categorical data were analyzed with the χ^2^ or Fisher exact test. A *p* value < 0.05 was considered statistically significant. Multivariable backward stepwise logistic regression (Wald) was used to determine the odds ratio of achieved primary efficacy endpoints: relative stent expansion >80%. Model included all predictors with a *p* value of less than 0.1 and without a significant multicollinearity effect. The data are presented as odds ratios (OR) with 95% confidence intervals (95% CI). The statistical analysis was performed using MedCalc version 17.9.2 (MedCalc Software, Ostend, Belgium).

## 3. Results

### 3.1. Patients, Lesion Characteristics, and Procedural Data

The multicenter IVL-DRAGON Registry included 62 patients. The mean age was 69 ± 7.1 years, and 66.1% were men. Acute coronary syndrome was the presenting diagnosis in 32 (51.6%) of cases (10 (16.1%) unstable angina, 20 (32.3%) non-ST segment elevation myocardial infarction, 2 (3.2%) ST-elevation myocardial infarction) and chronic coronary syndrome was present in 30 (48.4%) patients. The baseline clinical characteristics are presented in Table 1. The median time since index PCI of target lesion was 12 (4–54) months. Twenty-eight (45.2%) patients had diabetes mellitus, 10 (16.1%) chronic kidney disease, and 16 (25.8%) peripheral artery disease. Twenty eight patients (45.2%) had recurrent in-stent restenosis likely caused by stent underexpansion and 18 (29%) patients had more than one stent layer. In more than half of the patients, the underexpanded stent was implanted within a segment with heavy calcifications visible in an angiogram. In most patients, IVL PCI was performed in the right coronary artery (50%). For predilatation, a non-compliant balloon was used in 57 (91.9%) patients, a very high-pressure balloon (OPN NC, Sis Medical, Frauenfeld, Switzerland) was used in 3 (4.9%) patients, and a semi-compliant balloon was used in 2 (3.2%) patients. Postdilatation was performed in 53 (85.5%) patients. We did not observe no-reflow during PCI; however, one patient (1.6%) had perforation, and one (1.6%) patient had dissection. The angiographic and procedural characteristics are summarized in Table 2.

### 3.2. Primary Efficacy Endpoint and Secondary Endpoint

The procedural success was achieved in 54 (72.6%) patients. The logistic regression model demonstrated that independent predictors of unsuccess stent expansion >80% after IVL-PCI were CKD (OR 0.53; 95% CI 0.30–0.93, *p* = 0.030) and more than one stent layer in treated segment (OR 0.71; 95% CI 0.5–0.98; *p* = 0.048). The secondary endpoint of DOCE occurred in one (1.6%) patient caused by cardiac death, and there was no TLR or TV-MI during 30 days of follow-up. At the follow-up of 324.5 ± 206.0 days, DOCE occurred in 3 (4.8%) patients, cardiac death in 2 (3.2%) patients, TV-MI in 2 (3.2%) patients, and TLR in 2 (3.2%) patients. A representative example of the effects of IVL is shown in Figure 1.

### 3.3. Quantitative Findings

A comparison of lesion characteristics before and after IVL is presented in Table 3. QCA was available in all lesions treated with IVL. The mean lesion length by QCA was 21.2 ± 11.2 mm. Stent underexpansion expressed as in-stent diameter stenosis decreased significantly after IVL from 58.5% (47.5–69.7) to 11.4% (5.8–20.7), *p* < 0.001. Percentage area stenosis at minimum lumen diameter decreased from 82.6 (72.4–90.8) to 21.5 (11.1–37.2), *p* < 0.001, post IVL (Figure 2). The percentage change of the lumen stenosis diameter before and after IVL was 44.0 ± 18.1%.

### 3.4. OCT and IVUS Data

OCT imaging was carried out in 15 (24.2%) patients. In one patient, the lesion was non-crossable for OCT probe and imaging was performed only post IVL. Lesion preparation with IVL led to an increase in stent expansion at minimal stent area from 37.5% (16.0–66.0) to 86.0% (69.2–90.7), *p* < 0.001. The minimal lumen area 1.9 mm^2^ (1.7–2.9) vs. 5.8 mm^2^ (5.5–8.0), *p* < 0.001, and stent areas 2.8 mm^2^ (1.8–4.0) vs. 6.4 mm^2^ (5.5–8.1), *p* < 0.001, significantly increased after IVL. The maximal calcium arc behind the stent revealed significant differences before and after IVL 277 (235–313)° vs. 207 (175–240)°, *p* = 0.004, and maximum calcium thickness was 0.7 (0.6–0.8) mm before vs. 0.6(0.5–0.7) mm after IVL, *p* = 0.003. The IVUS imaging was performed in 14 (22.6%) patients. In one patient, the lesion was non-crossable for the IVUS probe and was performed only post IVL. Lesion preparation with IVL led to an increase in stent expansion at minimal stent area from 57.0% (31.5–77.2) to 89.0% (85.0–92.0), *p* = 0.002. The minimal lumen and stent area revealed significant differences before and after IVL 2.5 mm^2^ (1.8–2.9) vs. 5.1 mm^2^ (3.8–8.3), *p* < 0.001 and 4.1 mm^2^ (2.7–5.0) vs. 8.3 mm^2^ (6.7–8.5), *p* < 0.001, respectively.

## 4. Discussion

To the best of our knowledge, this is the largest cohort study to date addressing the safety and efficacy of IVL in patients with coronary stent underexpansion. The current registry extends prior data on the utility of IVL in de novo calcified lesions [8,9,10,17]. Our results, for the first time, lend support for the use of IVL in the setting of stent underexpansion and suggest that this approach is safe and effective in improving lumen and stent areas. The IVL angioplasty has a unique mechanism of action exerted on calcium when compared with a conventional high-pressure balloon, cutting or scoring technologies, rotational atherectomy, orbital atherectomy, or laser coronary angioplasty. These methods are known to substantially exacerbate the PCI risk with the possibility of stent damage, dissection, perforation, and adverse events during PCI [5,6,7]. By contrast, IVL—as an atraumatic, balloon-based treatment technology—may reduce mechanical vessel trauma and thus avoid life-threatening complications. Furthermore, debulking atherectomy devices often generates atheromatous debris that might embolize, causing microcirculatory disturbances with resultant slow or no-reflow myocardial ischemia, or infarction, whereas fractured greater calcium fragments generated by IVL appear to remain in situ [18]. Indeed, we did not observe no-reflow in this study, which is in line with some previous observations from the coronary as well as peripheral IVL angioplasty studies [19]. On the other hand, the IVL catheter has worse deliverability; the balloon is bulky and stiff, making it hazardous and unsuitable for use in the distal calcified artery segments. However, Hill et al. [9] in the Disrupt CAD III study showed that successful IVL delivery was achieved in as many as 98.2% of procedures. The major theoretical benefit of IVL in the treatment of stent underexpansion is that it modifies calcified coronary plaque, leading to behind-stent calcium disintegration. However, theoretically there is risk of disruption of DES polymer in fresh implantation as well as indenting its struts affecting corrosion by IVL therapy. However, this should be reflected in stent failure during follow-up, which was not observed in our study. Other devices such as excimer laser coronary angioplasty may favorably modify calcium plaque and thereby expand the stent. Lee et al. [20] compared excimer laser coronary angioplasty to the standard therapy in ISR calcium lesions caused by stent underexpansion, demonstrating the effectiveness of excimer laser coronary angioplasty for treating ISR with underexpansion by disrupting persistent calcium and facilitating better expansion of the previously implanted stent. The intravascular imaging data have supported our QCA observations. IVUS or OCT—performed to more accurately characterize the extent of calcification and provide insights into the mechanism of IVL in facilitating stent expansion—confirmed an evident increase in arterial luminal area. The acute procedural and clinical success rates of IVL angioplasty in underexpanded stent were found to be very promising, suggesting that IVL may represent the preferred modality in case of PCI with stent underexpansion. Still, longer-term follow-up is needed to understand how these acute results, particularly optimized stent expansion and minimal stent area, will translate into late clinical outcomes.

## 5. Study Limitations

Several limitations of the presented investigation need to be considered. First, the IVL-DRAGON Registry was a retrospective cohort study. Although consecutive patients treated with IVL were screened for eligibility, some selection bias cannot be excluded. Second, the presented findings warrant confirmation in a larger, prospective study with long-term clinical outcomes adjudication. Finally, the intracoronary imaging was not systematically performed in the overall study population.

## 6. Conclusions

In this real-life, largest-to-date analysis of IVL use to manage underexpanded stent, IVL presented as an effective and safe modality to facilitate stent expansion and luminal gain. Our findings warrant a larger, prospective study with long-term clinical outcomes adjudication to confirm IVL use as an emerging first-line therapeutic option to treat stent underexpansion.

## Figures and Tables

**Figure 1 jcm-11-01779-f001:**
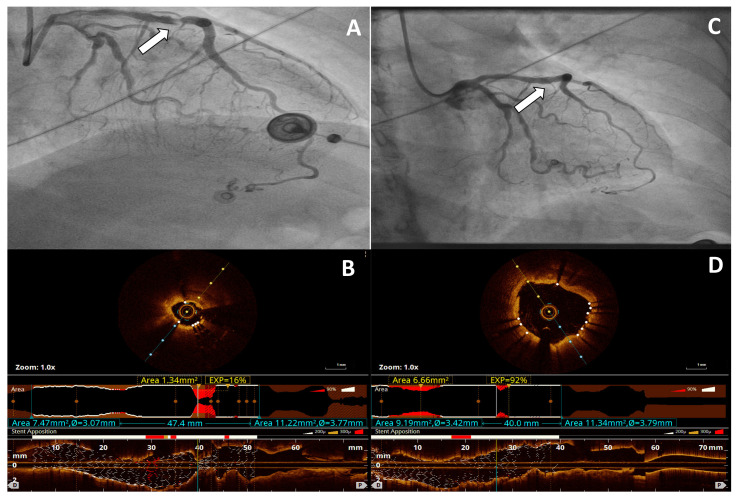
Representative example of angiographic and optical coherence tomography (OCT) images of Shockwave Intravascular Lithotripsy (IVL) for stent underexpansion caused by severe coronary artery calcification. (**A**) Angiography demonstrates significant lesion (arrow) in the left anterior descending artery caused by underexpanded stent. (**B**) OCT cross-sectional and longitudinal image acquired before IVL demonstrates deep calcium fractures and 16% stent expansion (**C**) Angiography demonstrates improvement in the area of stenosis (arrow) after IVL. (**D**) OCT cross-sectional and longitudinal image acquired post-IVL demonstrates 92% stent expansion. ‘D’ = distal; ‘P’ = proximal.

**Figure 2 jcm-11-01779-f002:**
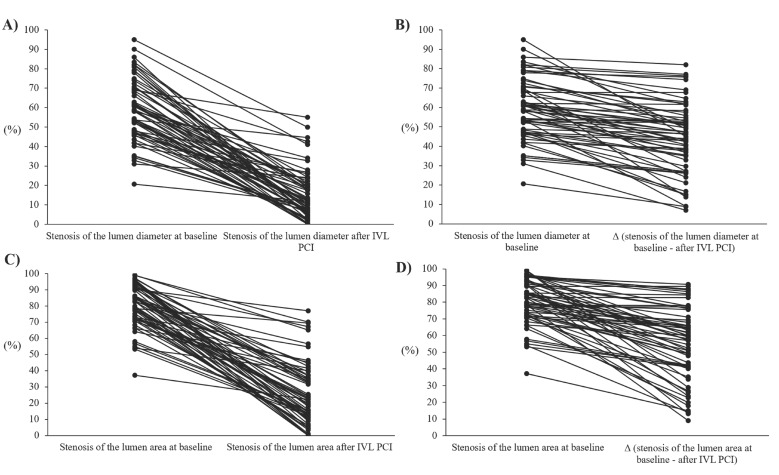
Cumulative frequency distribution curves demonstrating increased lumen diameter and lumen area at baseline and post Shockwave Intravascular Lithotripsy (IVL) percutaneous coronary intervention (PCI) calculated by quantitative coronary angiography (QCA). (**A**,**B**) Stenosis of the lumen diameter (**C**,**D**) Stenosis of the lumen area.

**Table 1 jcm-11-01779-t001:** Patient characteristics, risk factors, and clinical presentation.

	PCI with IVL
Age, y	69 ± 7.1
Male sex	41 (66.1)
Body mass index, kg/m^2^	30 ± 4.2
Discharge diagnosis	
Chronic coronary syndrome, n (%)	30 (48.4)
Unstable angina, n (%)	10 (16.1)
Non–ST-segment elevation myocardial infarction	20 (32.3)
ST-segment elevation myocardial infarction	2 (3.2)
Previous myocardial infarction	43 (69.4)
Previous CABG	10 (16.1)
Diabetes mellitus	28 (45.2)
On insulin	9 (14.5)
Hypertension	58 (93.5)
Hyperlipidemia	58 (93.5)
Chronic kidney disease *	10 (16.1)
Dialysis	1 (1.6)
Atrial fibrillation	15 (24.2)
Prior smoker	14 (22.6)
Current smoker	11 (17.7)
Pulmonary disease	10 (16.1)
Peripheral artery disease	16 (25.8)
Time since last PCI of target lesion, months	12.0 (4.0–54.0)
Recurrent in-stent restenosis	28 (45.2)
Number of in-stent restenosis events	1.8 ± 1.4
Left ventricular ejection fraction, %	48 ± 11.5

Values are mean ± standard deviation, n (%), or median (interquartile range). * Estimated glomerular filtration rate of <60 mL/min/1.73 m^2^ calculated using the Modification of Diet in Renal Disease method. CABG = coronary artery bypass grafting; PCI = percutaneous coronary intervention.

**Table 2 jcm-11-01779-t002:** Angiographic, procedural, and medication data.

	PCI with IVL (n = 62)
Number of diseased vessels	
1	30 (48.4)
2	15 (24.2)
3	17 (27.4)
Treated artery	
LM	6 (9.7)
LAD	16 (25.8)
LCx	9 (14.5)
RCA	31 (50.0)
Underexpansion in drug-eluting stent	61 (98.4)
Underexpansion in bare-metal stents	1 (1.6)
Number of stent layers	1.4 ± 0.8
Bifurcation lesion	9 (14.5)
Severe calcification on angiography	37 (59.7)
IVL data	
Total procedure time, min	86.3 ± 58.4
Fluoroscopy time, min	21.7 ± 13.2
Number of IVL catheters	1.0 ± 0.1
Number of pulses	66.3 ± 25.7
IVL balloon size, mm	3.3 ± 0.4
IVL balloon pressure, atm	8.0 ± 3.3
PCI with drug-eluting stent	24 (38.7)
PCI with drug-coated balloon	20 (32.3)
Plain old balloon angioplasty	18 (29.0)
Predilation	
Semi-compliant balloon	2 (3.2)
Non- compliant balloon	57 (91.9)
Very high-pressure balloon	3 (4.9)
Predilation balloon nominal diameter, mm	3.3 ± 0.5
Maximum predilation inflation pressure, atm	21.4 ± 5.2
Postdilation	
Semi-compliant balloon	4 (6.5)
Non-compliant balloon	49 (79.0)
Very high-pressure balloon	-
Postdilation balloon nominal diameter, mm	3.8 ± 0.5
Maximum postdilation inflation pressure, atm	19.6 ± 5.1
Complications	
Perforation	1 (1.6)
Dissection	1 (1.6)
No reflow	-
Procedural use of intracoronary imaging	
1-IVUS	14 (22.6)
2-OCT	15 (24.2)

Values are mean ± standard deviation and n (%); Cx = circumflex artery; LAD = left anterior descending; LM = left main; RCA = right coronary artery; OCT = optical coherence tomography; PCI = percutaneous coronary intervention; IVUS = intravascular ultrasound. IVL = intravascular lithotripsy.

**Table 3 jcm-11-01779-t003:** Quantitative findings and intracoronary imaging.

	Before IVL	Post IVL	*p*
Quantitative findings, n = 62 (100)			
Diameter stenosis at MLD, (%)	58.5 (47.5–69.7)	11.4 (5.8–20.7)	<0.001
Area stenosis at MLD, (%)	82.6 (72.4–90.8)	21.5 (11.1–37.2)	<0.001
MLD, (mm)	1.1(0.7–1.4)	2.6 (2.3–3.1)	<0.001
Lesion length, mm	21.2 ± 11.2	-	
OCT, n = 15 (24.2)			
Stent expansion at MSA, %	37.5 (16.0–66.0)	86.0 (69.2–90.7)	<0.001
MLA, mm^2^	1.9 (1.7–2.9)	5.8 (5.5–8.0)	<0.001
MSA, mm^2^	2.8 (1.8–4.0)	6.4 (5.5–8.1)	<0.001
Maximal calcium angle behind stent, °	277 (235–313)	207 (175–240)	0.004
Maximum calcium thickness, mm	0.7 (0.6–0.8)	0.6 (0.5–0.7)	0.003
Calcium length behind stent, mm	12.3 (11.1–17.3)	-	-
Malapposition,	4 (28.6)	1 (7.1)	0.146
Thrombus, n (%)	-	-	
Stent fracture, n (%)	-	-	
Tissue protrusion, n (%)	-	-	
Dissection/intramural hematoma, n (%)	-	-	
IVUS, n = 14 (22.6)			
Stent expansion at MSA, %	57.0 (31.5–77.2)	89.0 (85.0–92.0)	0.002
MLA, mm^2^	2.5 (1.8–2.9)	5.1 (3.8–8.3)	<0.001
MSA, mm^2^	4.1 (2.7–5.0)	8.3 (6.7–8.5)	<0.001
Plaque burden behind stent at MLA, %	71.0 (55.0–82.4)	40.0 (32.2–64.7)	0.051
Remodeling index	0.9 (0.8–0.9)	1.2 (1.0–1.4)	0.051
Dissection/intramural hematoma, n (%)	-	-	

Values are median (interquartile range) and n (%); OCT = optical coherence tomography; IVUS = intravascular ultrasound; MLD = minimum lumen diameter; MLA = minimum lumen area; MSA = minimum stent area.

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
