# Peer review of "Intravascular Lithotripsy for the Treatment of Stent Underexpansion: The Multicenter IVL-DRAGON Registry"

_jcm, 2022, doi:10.3390/jcm11071779_

Round 1

Reviewer 1 Report

In the manuscript "Intravascular Lithotripsy for the Treatment of Stent Underexpansion: the Multicenter IVL-DRAGON Registry", Wanha et al examine 62 patients who underwent IVL for stent underexpansion and found that >80% stent expansion by QCA was achieved in 73% of patients, confirmed internally by IVUS and OCT when performed. A few questions:

1) Do you have a comparator group - ie those undergoing ELCA or POBA as the pre-treatment? 

2) Please provide additional procedural data - guide size, radial versus femoral, ability to deliver IVL, what percentage received stent after IVL. 

3) in table 3 please include number of patients undergoing OCD and IVUS pre and post IVL. 

4) in the discussion, please describe other methods of managing stent under-expansion - ELCA, rotablation and compare how these may differ from the IVL mechanism. 

Author Response

Dear Editor,

Thank you for your response to the submission of our manuscript entitled: “Intravascular Lithotripsy for the Treatment of Stent Underexpansion: the Multicenter IVL-DRAGON Registry”.

 We appreciate the time and effort taken by reviewers to analyse our work and are thankful for the helpful criticism offered.

 We have carefully read the reviewers’ comments; have edited our manuscript to reflect the changes resulting from the review process.

 All the corrections in the manuscript were marked. Please do not hesitate to contact me directly if there are any further questions.

Sincerely,

Wojciech Wańha

In the manuscript "Intravascular Lithotripsy for the Treatment of Stent Underexpansion: the Multicenter IVL-DRAGON Registry", Wanha et al examine 62 patients who underwent IVL for stent underexpansion and found that >80% stent expansion by QCA was achieved in 73% of patients, confirmed internally by IVUS and OCT when performed. A few questions:

  • Do you have a comparator group - ie those undergoing ELCA or POBA as the pre-treatment? 

Response: We thank the Reviewer for this important comment. All patients had POBA before IVL (Table 2). We did not compare IVL with ELCA due to lack of ELCA control group.

  • Please provide additional procedural data - guide size, radial versus femoral, ability to deliver IVL, what percentage received stent after IVL. 

Response: We thank the Reviewer for the possibility to clarify this point. We have added details regarding treatment post IVL accordance to the Reviewer’s suggestions. Stent implantation was performed in 24 (38.7%) patients and drug-coated balloon was used in 20 (32.3%). (Table 2). Unfortunately, we do not have data of the guide size and vascular access.

  • In table 3 please include number of patients undergoing OCT and IVUS pre and post IVL. 

Response: We have now added details accordance to the Reviewer’s suggestions. (Table 3).

  • In the discussion, please describe other methods of managing stent under-expansion - ELCA, rotablation and compare how these may differ from the IVL mechanism. 

Response: We have described other methods of managing stent under-expansion in the discussion accordance to the Reviewer’s suggestions.

Reviewer 2 Report

Authors should provide some information more information on the decision making process to treat an angiographically underexpanded stent.  Was hemodynamic assessment performed?  Since intravascular imaging was used in less than 25% of cases, was the decision to refer for IVL purely based on QCA?

The authors should include some discussion about theoretical disruption of DES polymer by IVL therapy and implications for drug elution and/or long  term stent patency. 

Author Response

Dear Editor,

Thank you for your response to the submission of our manuscript entitled: “Intravascular Lithotripsy for the Treatment of Stent Underexpansion: the Multicenter IVL-DRAGON Registry”.

 We appreciate the time and effort taken by reviewers to analyse our work and are thankful for the helpful criticism offered.

 We have carefully read the reviewers’ comments; have edited our manuscript to reflect the changes resulting from the review process.

 All the corrections in the manuscript were marked. Please do not hesitate to contact me directly if there are any further questions.

 Sincerely,

 Wojciech Wańha

Authors should provide some information more information on the decision making process to treat an angiographically underexpanded stent.  Was hemodynamic assessment performed?  Since intravascular imaging was used in less than 25% of cases, was the decision to refer for IVL purely based on QCA?

Response: We thank the Reviewer for this important comment. According decision-making process we add that information in the discussion and conclusion section. The acute procedural and clinical success rates of IVL angioplasty in underexpanded stent were found to be very promising, suggesting that IVL may represent the preferred modality in case of PCI with stent underexpansion. Our findings warrant a larger, prospective study with long-term clinical outcomes adjudication to confirm IVL use as an emerging first-line therapeutic option to treat stent underexpansion. The hemodynamic assessment was performed only if needed due to complications which were in one patient.

The authors should include some discussion about theoretical disruption of DES polymer by IVL therapy and implications for drug elution and/or long term stent patency. 

Response: We thank the Reviewer for this important and interesting comment. Potentially there is risk of theoretical disruption of DES polymer in fresh implantation as well as indenting its metal—predisposing corrosion by IVL therapy. But this should be reflected in stent failure during follow-up what was not observed in our study (discussion).